# Sero-prevalence of 19 infectious pathogens and associated factors among middle-aged and elderly Chinese adults: a cross-sectional study

Pang Yao ,[1] Iona Millwood,[1,2] Christiana Kartsonaki,[1,2] Alexander J Mentzer,[3] Naomi Allen,[1] Rima Jeske ,[4] Julia Butt,[4] Yu Guo,[5] Yiping Chen,[1,2] Robin Walters,[1,2] Jun Lv ,[6] Canqing Yu,[6] Martyn Plummer,[7] Catherine de Martel,[8] Gary Clifford,[8] Li-ming LI,[6,9] Tim Waterboer,[4] Ling Yang,[1,2] Zhengming Chen[1]

TW, LY and ZC are joint senior authors.

**Correspondence to**
Dr Ling Yang;
ling.yang@ndph.ox.ac.uk

## ABSTRACT

**Objectives** To systematically assess the sero-prevalence and associated factors of major infectious pathogens in China, where there are high incidence rates of certain infection-related cancers.

**Design** Cross-sectional study.

**Setting** 10 (5 urban, 5 rural) geographically diverse areas in China.

**Participants** A subcohort of 2000 participants from the China Kadoorie Biobank.

**Primary measures** Sero-prevalence of 19 pathogens using a custom-designed multiplex serology panel and associated factors.

**Results** Of the 19 pathogens investigated, the mean number of sero-positive pathogens was 9.4 (SD 1.7), with 24.4% of participants being sero-positive for >10 pathogens. For individual pathogens, the sero-prevalence varied, being for example, 0.05% for HIV, 6.4% for human papillomavirus (HPV)-16, 53.5% for *Helicobacter pylori* (*H. pylori*) and 99.8% for Epstein-Barr virus . The sero-prevalence of human herpesviruses (HHV)-6, HHV-7 and HPV-16 was higher in women than men. Several pathogens showed a decreasing trend in sero-prevalence by birth cohort, including hepatitis B virus (HBV) (51.6% vs 38.7% in those born <1940 vs >1970), HPV-16 (11.4% vs 5.4%), HHV-2 (15.1% vs 8.1%), *Chlamydia trachomatis* (65.6% vs 28.8%) and *Toxoplasma gondii* (22.0% vs 9.0%). Across the 10 study areas, sero-prevalence varied twofold to fourfold for HBV (22.5% to 60.7%), HPV-16 (3.4% to 10.9%), *H. pylori* (16.2% to 71.1%) and *C. trachomatis* (32.5% to 66.5%). Participants with chronic liver diseases had >7-fold higher sero-positivity for HBV (OR=7.51; 95% CI 2.55 to 22.13).

**Conclusions** Among Chinese adults, previous and current infections with certain pathogens were common and varied by area, sex and birth cohort. These infections may contribute to the burden of certain cancers and other non-communicable chronic diseases.

## INTRODUCTION

Chronic infections with certain pathogens, including viruses, bacteria and parasites, have been associated with increased risks of several non-communicable chronic diseases

(NCDs), especially certain cancers.[1] In 2018, an estimated 2.2 million cancer cases worldwide were attributed to infections, mainly *Helicobacter pylori* (8 10 000 cases), human papillomavirus (HPV; 690 000), hepatitis B virus (HBV; 360 000) and hepatitis C virus (HCV; 160 000).[1] The incidence rates of infection-related cancers were highest in eastern Asia, particularly China, owing to the high prevalence of *H. pylori* and HBV.[1] Apart from cancer, a growing body of evidence also suggests that inflammation induced by persistent pathogen infection may be associated with risks of cardiovascular disease (cytomegalovirus, CMV),[2] multiple scleroses (Epstein-Barr virus (EBV))[3] and Alzheimer's disease (human herpesviruses (HHV) 6 and 7).[4] Moreover, there is also evidence suggesting that pathogen burden (ie, the number of different infectious pathogens an individual has been exposed to) may contribute to atherosclerosis and risk of cardiovascular disease.[2]

The prevalence of pathogens tends to vary greatly between and within populations and is often associated with disparities in socioeconomic status, household crowding, breastfeeding practices, food production practices and environmental factors such as climate.[5] Although many sero-epidemiological studies of major infectious pathogens have been undertaken,[6–9] they have tended to cover a single pathogen[6–8] or involve populations in high-income countries.[9] In China, although several nationwide surveys have previously reported on the prevalence of some major pathogens (eg, HBV, HCV),[10–12] there is limited contemporary data about the sero-prevalence of many other pathogens and their relevance for major NCDs beyond certain cancers. Moreover, few studies in China or elsewhere have systematically examined the sero-prevalence of multiple pathogens in the same population, which could provide important data about pathogen burden and the joint effects of different pathogens on disease aetiology.

The present study examined the sero-prevalence of 19 infectious pathogens and investigated certain demographic factors associated with sero-positivity among 2000 adults from the China Kadoorie Biobank (CKB) study that covered 10 geographically diverse urban and rural areas in China.

## MATERIALS AND METHODS
### Study population
Details of the CKB study design, methods and participants have been previously reported.[13] The 512 715 participants aged 30–79 years were enrolled from 10 (5 urban, 5 rural) geographically diverse regions in China (online supplemental figure 1). At baseline (June 2004–July 2008) and at two subsequent resurveys in a 5% subset (in 2008 and 2013, respectively), detailed data were collected on sociodemographic characteristics, smoking, alcohol consumption, diet, physical activity, personal and family medical history and physical measurements. Follow-up of CKB participants was through linkage to established mortality and morbidity registries and to the nationwide health insurance system, which records all hospitalised episodes The present study population was a subcohort of 2000 participants from a case-subcohort study of gastric and oesophageal cancers in CKB. This subcohort was selected using simple random sampling from those alive with no history of cancer 2 years after entering the study who had an available plasma sample collected at baseline and had genotyping data available as part of a population-based subset.

### Patient and public involvement
No patients or population groups were involved.

### Blood sample collection and processing
At baseline, a 10 mL non-fasting venous blood sample was collected from each participant into an EDTA vacutainer (with time of last meal recorded). Samples were separated into plasma and buffy coat aliquots and frozen within 24 hours of collection (mean delay ~10 hours). Two aliquots of plasma from each blood sample were shipped to Oxford, UK for long-term storage in liquid nitrogen following the formal approval of regulatory agencies in China.

### Multiplex serology
The multiplex serology panel used in the present study was based on that developed for the UK Biobank,[9] with certain relevant modifications. More *H. pylori* antigens were included in the CKB panel and a few pathogens were removed taking their prevalence into account.[9] The CKB panel measured antibody levels against 43 antigens from 19 infectious pathogens (see details in online supplemental table 1) including hepatitis B (HBV) and hepatitis C (HCV) viruses; HPV-16 HPV-18; herpes simplex viruses 1 and 2 (HHV-1 and HHV-2); varicella zoster virus (VZV); EBV; human CMV; HHV-6 and HHV-7; BK (BKV), JC (JCV) and Merkel cell (MCV) polyomaviruses; HIV; human T lymphotropic virus (HTLV); *H. pylori; Chlamydia trachomatis* and *Toxoplasma gondii*. Of these 19 pathogens, 8 (HBV, HCV, HIV, HTLV, *H. pylori*, HPV-16, HPV-18 and EBV) are classified as group I human carcinogens by the International Agency for Research on Cancer (IARC).

The quantitative multiplex antibody detection approach was based on a glutathione S-transferase (GST) capture ELISA combined with fluorescent-bead technology, with various validation procedures.[14] Antigens of infectious pathogens were bacterially expressed as GST-X-tag fusion proteins and coupled to spectrally distinguishable glutathione–casein covered beads. The median fluorescence intensity (MFI) of at least 100 beads per antigen was measured, and, thus, a sample was defined sero-positive for specific antibody against a corresponding antigen if the respective MFI value exceeded the antigen-specific cut-off. Antigen-specific cut-offs for *H. pylori* antigens were applied as described previously[15] and were quality assured by the visual inflection point method.[16] To define overall sero-positivity for a pathogen, antigen-specific sero-positivity results were subsequently combined using published algorithms (see details in online supplemental table 1 and figure 2).[9]

For 100 (5%) of these participants, plasma samples collected at the two subsequent resurveys were also assayed to assess the consistency of sero-status (sero-conversion and sero-reversion rates) measurements over a period of time.

### Statistical analysis
Sero-prevalence estimates and 95% CIs for each antigen and pathogen were calculated. For pathogens, sero-prevalence was also estimated in subgroups defined by sex, birth cohort (each year) and study area (10). Locally weighted scatterplot smoothing curves were used to visualise the relationship between birth cohort and sero-prevalence of each pathogen, and proportion of individuals with >10 infectious pathogens or >2

oncogenic pathogens. Logistic regression was used to estimate ORs by pathogen sero-positivity for various baseline characteristics, adjusting for age, sex and area. Pathogens with extreme sero-prevalence (ie, <5% or >95%) were excluded from the association analyses. Continuous MFI values were used to compute Pearson correlation coefficients between each of the antigens. Spearman's correlation coefficients were calculated between pathogens. The analysis was done using R V.3.6.2.

## RESULTS
### Characteristics of the study population
Among the subcohort of 1986 participants (14 samples were excluded due to technical errors) included in the main analyses, the mean (SD) age was 51.8 (10.8) years, 61.7% were women and 50.3% were urban residents, which, along with many other baseline characteristics, were similar to those in the overall CKB cohort (table 1).

## SERO-PREVALENCE OF PATHOGENS
Of the 19 pathogens investigated, the sero-prevalence estimates ranged from 0.05% to 99.8% for individual pathogens (table 2). Within specific families of pathogens, the sero-prevalences also varied greatly. For the hepatitis viruses, the sero-prevalence was 43.8% for HBV and 0.7% for HCV. Among papillomaviruses, HPV-16 sero-prevalence was higher (6.4%) than HPV-18 (3.4%). The sero-prevalence of herpesviruses was 98.0% for HHV-1, 8.9% for HHV-2, 91.5% for VZV, 99.8% for EBV, 97.3% for CMV, 62.0% for HHV-6, and 85.5% for HHV-7. Among the polyomaviruses, BKV had the highest sero-prevalence (91.1%), followed by JCV (65.6%) and MCV (64.9%). For HIV and HTLV, however, only samples from 1 (0.05%) and 2 (0.1%) participants tested positive, respectively. The sero-prevalences of *H. pylori, C. trachomatis* and *T. gondii* were 53.5%, 47.6% and 14.2%, respectively.

No differences by sex were observed for most pathogens but, compared with men, women had a higher sero-prevalence of HPV-16 (7.9% vs 4.1%), HHV-6 (68.0% vs 52.3%) and HHV-7 (89.3% vs 79.5%). A decreasing trend in sero-prevalence by birth cohort was seen for HBV (51.6% in those born before 1940 to 38.7% in those born after 1970), HPV-16 (11.4% to 5.4%), HHV-2 (15.1% to 8.1%), JCV (75.5% to 61.3%), MCV (73.6% to 62.2%), *C. trachomatis* (65.6% to 28.8%) and *T. gondii* (22.0% to 9.0%), while no clear trends were evident for other pathogens, except HHV-6, which showed an upward trend (figure 1). Across the 10 study areas, there were at least twofold differences in sero-prevalence for HBV (22.5% to 60.7%), HPV-16 (3.4% to 10.9%), HHV-2 (4.3% to 17.8%), *H. pylori* (16.2% to 71.1%), *C. trachomatis* (32.5% to 66.5%) and *T. gondii* (8.0% to 18.4%) (figure 2).

Among the 1986 participants and 19 pathogens tested, all participants were sero-positive for at least 1 pathogen and 24.4% were sero-positive for more than 10 pathogens, with the mean number being 9.4 (SD, 1.7) (figure 3). The

mean number of oncogenic pathogens for which a participant was sero-positive was 2.1 (SD, 0.8) out of eight oncogenic pathogens. Overall, there was a clear decreasing trend by birth cohort in the proportions of participants sero-positive for >10 pathogens or >2 oncogenic pathogens (figure 3).

**Table 1** Baseline characteristics of the study subcohort and overall CKB cohort

| Characteristics | Subcohort (n=1986) | CKB (n=453 157) |
|---|---|---|
| Demographic and lifestyle factors | | |
| Age, years | 51.8 (10.8) | 51.7 (10.6) |
| Birth year, % | | |
| ≤1939 | 12.0 | 11.0 |
| 1940–1949 | 20.0 | 20.6 |
| 1950–1959 | 33.2 | 31.9 |
| 1960–1969 | 29.2 | 30.8 |
| ≥1970 | 5.6 | 5.7 |
| Female, % | 61.7 | 60.2 |
| Urban residents, % | 50.3 | 45.4 |
| Education ≥high school, % | 21.9 | 21.5 |
| Household income (>20 000 yuan/year), % | 42.4 | 43.2 |
| Living alone, % | 3.6 | 2.8 |
| Ever regular smoker in men, % | 73.9 | 74.0 |
| Ever regular smoker in women, % | 3.6 | 3.1 |
| Ever regular drinker in men, % | 38.4 | 36.9 |
| Ever regular drinker in women, % | 3.2 | 2.4 |
| Physical activity, MET h/day | 21.1 (14.2) | 21.3 (13.9) |
| Medical history and health status, % | | |
| Poor self-rated health | 10.2 | 10.0 |
| HBsAg positive* | 3.2 | 3.1 |
| Diabetes† | 5.7 | 5.6 |
| History of peptic ulcer | 3.9 | 3.9 |
| History of cirrhosis/chronic hepatitis | 1.3 | 1.2 |
| Family history of cancer | 14.8 | 14.2 |
| Anthropometry, mean (SD) | | |
| BMI, kg/m² | 23.8 (3.5) | 23.6 (3.4) |
| Waist circumference, cm | 80.1 (9.9) | 80.1 (9.7) |
| SBP, mm Hg | 130 (21) | 130 (21) |
| Random plasma glucose, mmol/L | 5.8 (2.3) | 6.0 (2.3) |

The results are presented as means (SD) or percentages.

*HBsAg was tested using an on-site rapid test strip at baseline.
†Includes those with a (self-reported) previous medical diagnosis of diabetes and those detected by blood glucose tests at baseline.
BMI, body mass index; CKB, China Kadoorie Biobank; HBsAg, Hepatitis B surface antigen; MET-h, metabolic equivalent of task-hours; SBP, systolic blood pressure.

**Table 2** Sero-prevalence estimates (95% CI) for infectious pathogens in men and women

| Pathogens | Men (n=761) | Women (n=1225) | All (n=1986) |
|---|---|---|---|
| Hepatitis viruses | | | |
| HBV | 46.1 (42.5 to 49.7) | 42.4 (39.6 to 45.2) | 43.8 (41.6 to 46.0) |
| HCV | 0.5 (0.1 to 1.3) | 0.8 (0.4 to 1.5) | 0.7 (0.4 to 1.2) |
| Human papillomaviruses (HPV) | | | |
| HPV-16 | **4.1 (2.8 to 5.7)** | **7.9 (6.5 to 9.6)** | 6.4 (5.4 to 7.6) |
| HPV-18 | 2.8 (1.7 to 4.2) | 3.8 (2.8 to 5.0) | 3.4 (2.6 to 4.3) |
| Human herpesviruses (HHV) | | | |
| HHV-1 | 97.2 (95.8 to 98.3) | 98.4 (97.6 to 99.1) | 98.0 (97.3 to 98.6) |
| HHV-2 | 7.1 (5.4 to 9.2) | 10.0 (8.3 to 11.8) | 8.9 (7.6 to 10.2) |
| VZV | 92.9 (90.8 to 94.6) | 90.7 (88.9 to 92.3) | 91.5 (90.2 to 92.7) |
| EBV | 99.9 (99.3 to 100.0) | 99.8 (99.4 to 100.0) | 99.8 (99.6 to 100.0) |
| CMV | 95.9 (94.3 to 97.2) | 98.1 (97.2 to 98.8) | 97.3 (96.5 to 98.0) |
| HHV-6 | **52.3 (48.7 to 55.9)** | **68.0 (65.3 to 70.6)** | 62.0 (59.8 to 64.1) |
| HHV-7 | **79.5 (76.5 to 82.3)** | **89.3 (87.4 to 91.0)** | 85.5 (83.9 to 87.1) |
| Human polyomaviruses (HPyV) | | | |
| BKV | 90.8 (88.5 to 92.8) | 91.3 (89.6 to 92.9) | 91.1 (89.8 to 92.4) |
| JCV | 68.5 (65.0 to 71.8) | 63.8 (61.1 to 66.5) | 65.6 (63.5 to 67.7) |
| MCV | 64.7 (61.1 to 68.1) | 65.1 (62.3 to 67.7) | 64.9 (62.8 to 67.0) |
| Other viruses | | | |
| HIV | 0 | 0.1 (0.0 to 0.5) | 0.1 (0.0 to 0.3) |
| HTLV | 0.1 (0.0 to 0.7) | 0.1 (0.0 to 0.5) | 0.1 (0.0 to 0.4) |
| Bacteria/parasite | | | |
| *H. pylori* | 54.3 (50.7 to 57.9) | 53.0 (50.1 to 55.8) | 53.5 (51.3 to 55.7) |
| *C. trachomatis* | 45.7 (42.1 to 49.3) | 48.8 (46.0 to 51.7) | 47.6 (45.4 to 49.9) |
| *T. gondii* | 12.4 (10.1 to 14.9) | 15.3 (13.4 to 17.5) | 14.2 (12.7 to 15.8) |

Bold values denote statistical significance at the p<0.05 level.
BKV, BK virus; CMV, cytomegalovirus; EBV, Epstein-Barr virus; HBV, hepatitis B virus; HCV, hepatitis C virus; HTLV, human T lymphotropic virus; JCV, JC virus; MCV, Merkel cell virus; VZV, varicella zoster virus.

## Factors associated with pathogen sero-positivity

The associations between sero-positivity of three major oncogenic pathogens (*H. pylori*, HPV-16 and HBV) and demographic and lifestyle factors are shown in online supplemental table 2. Sero-positivity for HBV was more common in older participants ($P_{trend}$=0.05), those with cirrhosis or chronic hepatitis (adjusted OR=7.51, 2.55 to 22.13), and those who were hepatitis B surface antigen (HBsAg; measured using a point-of-care test) positive (11.51, 5.16 to 25.7) at baseline. The factors significantly associated with HPV-16 infection were older age ($P_{trend}$=0.02), being women (2.07, 1.36 to 3.14), higher levels of education (1.63, 1.01 to 2.65) or low levels of physical activity (0.50, 0.32 to 0.78), having diabetes (2.69, 1.52 to 4.77) and obesity/overweight status (1.51, 1.03 to 2.22). For *H. pylori*, urban residence was associated with higher sero-prevalence (1.80, 1.50 to 2.15) and the converse was observed for a history of peptic ulcer (0.52, 0.32 to 0.85).

For the HHV family, women were more likely to be sero-positive for HHV-2 (1.45, 1.04 to 2.03), HHV-6 (1.92, 1.59 to 2.31) and HHV-7 (2.15, 1.67 to 2.76) than men. Higher physical activity levels were associated with lower likelihood of VZV sero-positivity while participants who reported self-rated poor health were less likely to be HHV-7 sero-positive. For JCV, diabetes and central obesity were associated with lower likelihood of sero-positivity, while for *C. trachomatis*, higher levels of education and income were associated with lower, and diabetes was associated with higher sero-prevalence.

## Correlation between different pathogens

Correlations between the 19 different pathogens were generally low (online supplemental figure 3). However, for certain pathogens, there were high correlations between their different antigens, for example, HBc and HBe (r=0.99; p<0.001) for HBV; core antigen and NS3 (r=0.95; p<0.001) for HCV, and HP1564 and CagA (r=0.60;

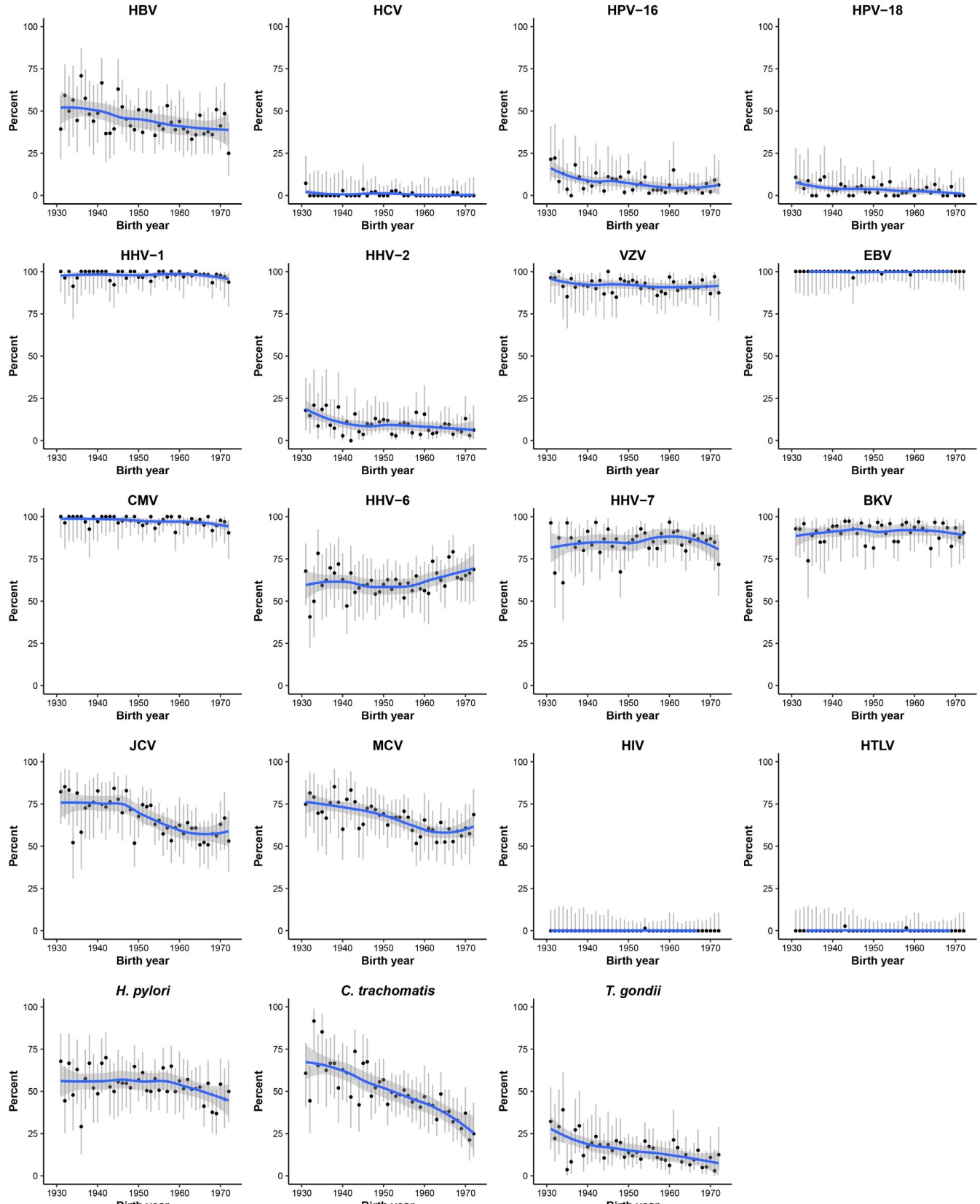

**Figure 1** Sero-prevalence of multiple pathogens by year of birth. Dots and grey lines represent sero-prevalence estimates and 95% CI for every single birth year. The blue line represents estimates for these sero-prevalences derived from LOESS method, and the shaded areas show the 95% CI. Due to few participants that were born before 1932 or after 1971, those who were born between 1927 and 1931, or 1971 and 1976 were combined, separately. BKV, BK virus; HBV, hepatitis B virus; HCV, hepatitis C virus; HHV, human herpesvirus; HPV, human papillomavirus; HTLV, human T lymphotropic virus; JCV, JC virus; MCV, Merkel cell virus; LOESS, locally weighted scatterplot smoothing.

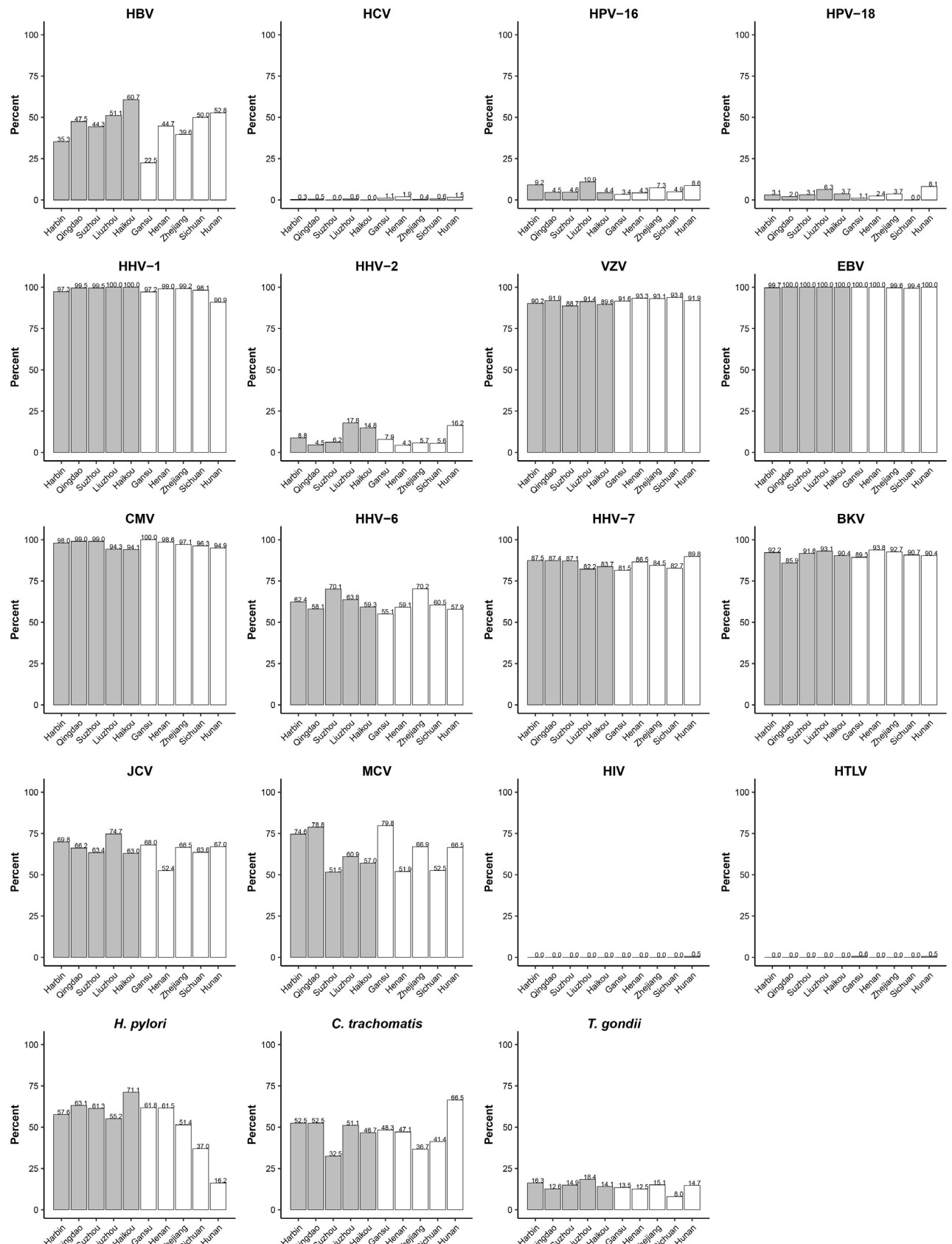

**Figure 2** Sero-prevalence of multiple pathogens across 10 study areas. The grey bar represents urban participants and white bar represents rural participants. The number of participants in each region was 198 (Qingdao), 295 (Harbin), 135 (Haikou), 194 (Suzhou), 174 (Liuzhou), 162 (Sichuan), 178 (Gansu), 208 (Henan), 245 (Zhejiang) and 197 (Hunan). BKV, BK virus; EBV, Epstein-Barr virus; HBV, hepatitis B virus; HCV, hepatitis C virus; HHV, human herpesvirus; HPV, human papillomavirus; HTLV, human T lymphotropic virus; JCV, JC virus; MCV, Merkel cell virus; VZV, varicella zoster virus.

## A Any pathogens

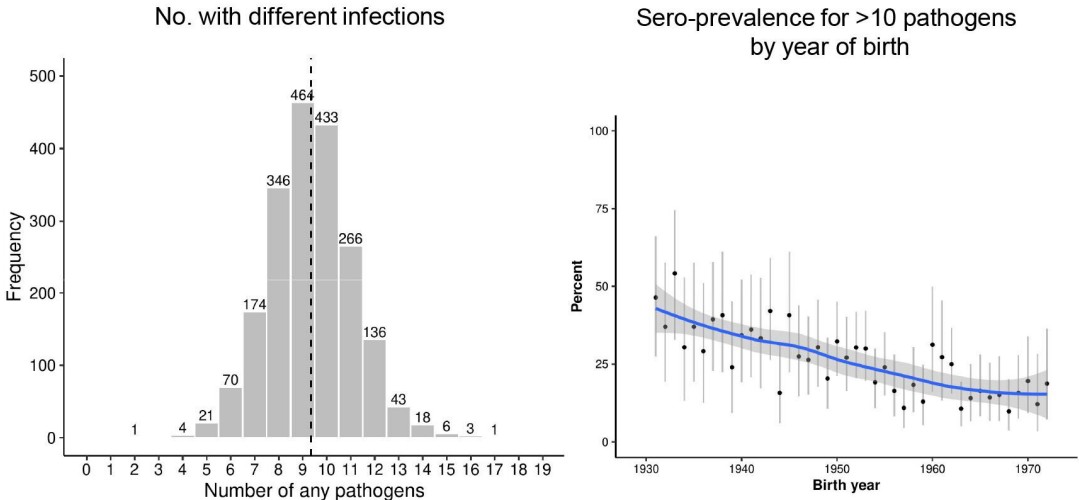

## B Oncogenic pathogens

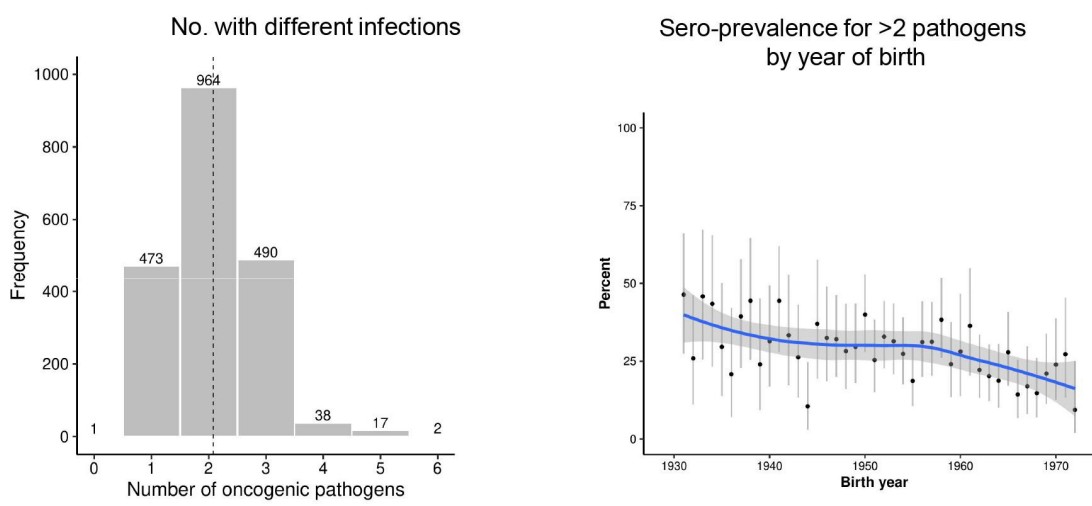

**Figure 3** Distribution of number of infections for (A, B) any of 19 pathogens; and (C, D) oncogenic pathogens, overall and by year of birth. Numbers with different infections (A and C), and sero-prevalence for >10 pathogens (B), or >2 pathogens (D) by birth years were presented. Pathogens of HBV, HCV, HIV, HTLV, *H. pylori*, HPV (16, 18) and EBV are classified as Group I human carcinogen by the IARC. The dashed lines represent the mean numbers of sero-positive pathogens among the study participants, which were 9.4 (SD 1.7) for any of 19 pathogens, and 2.1 (SD 0.8) for any of 8 oncogenic pathogens. Proportion of individuals with >10 infectious pathogens, or with >2 oncogenic pathogens was presented by birth cohort. Due to few participants that were born before 1932 or after 1971, those who were born between 1927 and 1931, or 1971 and 1976 were combined, separately. The blue line represents estimate for these sero-prevalence derived from LOESS method, and the shaded areas show the 95% CI. EBV, Epstein-Barr virus; HBV, hepatitis B virus; HCV, hepatitis C virus; HPV, human papillomavirus; HTLV, human T lymphotropic virus.

p<0.001) for *H. pylori* (online supplemental figure 4). For certain different pathogens, moderate correlations were also observed between certain antigens, such as HPV-16 L1 and *T. gondii* p22 (*r*=0.56; p<0.001).

### Repeat measurements

The MFI values were moderately to highly correlated between the baseline survey, and the first and second resurveys (online supplemental figure 5–7). Overall, the mean (SD) ICC for 43 antigens was 0.76 (0.18) between baseline and the first resurvey (mean SD duration of 2.5 (0.7) years). The apparent sero-conversion and

sero-reversion rates were generally low (ranging from 0% to 11.3% for different pathogens) across surveys (online supplemental table 3).

### DISCUSSION

In this large study of multiple pathogens in the general adult Chinese population, there were large differences in the sero-prevalence of different pathogens, both overall and by sex, birth cohort and areas. Moreover, certain socioeconomic factors (eg, levels of education

and income), lifestyle factors (eg, physical activity) and medical history (eg, having cirrhosis or chronic hepatitis) were also shown to be associated with sero-prevalence of many pathogens. Among the study population and the 19 pathogens tested, the mean number of sero-positive pathogens was more than 9.

The present study shows there is a substantial burden of past and current infections with various types of pathogens among Chinese adults. Compared with data from a similar study in the UK population involving about 10 000 adults (56% women),[9] certain infections were more prevalent in Chinese adults, particularly HBV, HHV-1, CMV, *H. pylori* and *C. trachomatis*, while others were less common, including HHV-2, HHV-6, HHV-7 and *T. gondii* (online supplemental table 4). Overall, the mean number of sero-positive pathogens was slightly greater in Chinese than in UK adults (any pathogen 9.4 vs 8.4; oncogenic pathogens: 2.1 vs 1.4), suggesting that the overall burden of pathogen infections might be generally higher in Chinese populations.

### HBV and HCV

For HBV, we measured antibody responses for HBc and HBe, which indicate past and current natural HBV infection, respectively, and the rate was approximately 44% in the present study, much higher than the 3% reported in the UK Biobank.[9] Our sero-prevalence estimate was, however, lower than recent data (54.7%) from a much smaller study (214 controls in a nested case–control study, samples obtained from 1996 to 2000) in China and Singapore using the same antigens (HBc and HBe) measured using multiplex serology.[17] Several Chinese nationwide surveys indicated that the prevalence of chronic HBV infection among adults measured by HBsAg had decreased steadily over the past two decades, from 10% in 1992,[10] to 7% in 2006,[11] and to 6% in 2010–2012.[12] In line with previous findings,[10–12] we also observed a declining sero-prevalence of HBV across birth cohorts, which was not influenced by the universal HBV vaccination programme that was implemented about 20 years ago and restricted to new borns. Our finding of a sero-prevalence of HCV (0.7%) is consistent with recent report from China using a similar assay (0.9%) based on a smaller sample size (n=214)[17] and higher than the 0.3% sero-prevalence in the UK Biobank.[9] Our study showed that HBV sero-positivity was >7-fold higher in participants who had reported prior cirrhosis or chronic hepatitis than in their counterparties. This is not surprising, since both HBV and HCV can lead to progressive liver cirrhosis, fibrosis and/or hepatocellular carcinoma and, thus, have been classified as group I human carcinogens for hepatocellular carcinoma.

### HPV-16 and HPV-18

The two most common HPVs, HPV-16 and HPV-18, are sexually transmitted oncogenic viruses, which account for 70%–75% of all cervical cancers and 40%–60% of cervical cancer precursors.[18] In addition to sex differences, HPV sero-prevalence may be affected by differences in socio-economic conditions, cultural diversity, assay methods and survey period. In the present study, the sero-prevalences of HPV-16 (overall 6.4%, urban 7.0%, rural 5.9%) and HPV-18 (overall 3.4%, urban 3.5%, rural 3.2%) were comparable to those reported in Chinese living in Taiwan (n=1702; 7.6% and 3.9%, respectively) measured using HPV virus-like particle ELISA,[19] but somewhat lower than those reported in another study of 3790 (median age of 44 years, 58% women) rural Chinese from Central China (9.9% and 8.9%, respectively) measured using multiplex serology.[20] A recent meta-analysis involving 24 Chinese provinces and autonomous regions indicated that HPV-16 and HPV-18 were more prevalent in Central and Northwest China than the rest of the country.[21] The increasing trend between HPV-16 sero-prevalence and age was consistent with previous findings,[7 19] which may reflect the fact that cumulative exposure measured by HPV sero-positivity is higher among older individuals and that antibodies to HPV-16 tend to be persistent.[7] As in the present study, many previous studies have consistently reported a higher sero-prevalence of HPV-16 in women than men.[19] It is possible that men are either not as susceptible to HPV-16 infection or more frequently able to clear the infection spontaneously without developing a systemic antibody response. Previous meta-analyses have reported that the overall HPV viral DNA prevalence among Chinese women ranged from 13.1% to 18.8%.[21] Targeted HPV vaccination and scale-up of cervical cancer screening for women are priorities in curbing the HPV epidemic in China.

### Human herpesviruses

Herpesviruses are a large family of enveloped DNA viruses that are ubiquitous in humans and characteristically establish lifelong latent infection within host cells and subsequently reactivate under certain conditions. Eight herpesvirus types are known to infect humans (HHV-1 to HHV-8), and both primary infection and reactivation can lead to a wide spectrum of clinical manifestations, such as encephalitis, congenital defects and cancer. EBV is causally associated with various types of lymphoma and nasopharyngeal cancer and has been classified as Group I human carcinogen by IARC.[22] This study shows that the age-standardised sero-prevalence of HHV-1 and CMV (HHV-5) is much higher in China (93.4% and 92.7%) than the UK (68.6% and 55.5%) or the USA (48.1% and 50.4%), while HHV-2 is lower in China (8.3%) than the UK (16.8%) or the USA (12.1%).[9] Similar ethnic differences were also observed within the UKB population,[9] with Asians (77.9% and 92.0%; n=236) having a higher sero-prevalence of HHV-1 and CMV than white participants (69.1% and 56.5%; n=9140). Moreover, we also observed fourfold (4.3%–17.8%) geographical differences for HHV-2 across the 10 areas in this study. Limited studies conducted in China have demonstrated HHV-2 sero-prevalence as high as 13.2% among 2141 rural residents (5–60 years) of Zhejiang province in 2006,[23] and as low as 3.4% among

a general population of 8074 residents (18–49 years) of Shandong province in 2016, with rates typically higher in women than men.[8] HHV-1, CMV and EBV are ubiquitous herpesviruses that usually are transmitted by contact with infected saliva or breast feeding, whereas HHV-2 is almost always transmitted through sexual contacts.[24] Ethnic and geographical differences in sero-prevalences might be explained by differences in behaviour that facilitate transmission.[25] For example, a multiethnic population-based prospective cohort study of 4464 Dutch children reported that lifestyle factors including low family net household income, low maternal educational level and crowding could explain up to 48% of the ethnic differences in HHV-1 sero-prevalences and up to 39% of the ethnic differences in EBV sero-prevalences.[25]

### Human polyomaviruses

Polyomaviruses are potentially tumorigenic in humans. The observed sero-prevalences of BKV (91.1%), JCV (65.6%) and MCV (64.9%) in the present study were similar to those reported previously among Chinese using multiplex serology (206 controls with mean age of 58 years: BKV 91.3%, JCV 73.3% and MCV 67.0%; 5548 participants with mean age of 61 years: MCV 61.0%).[26 27] Limited data exist on the individual characteristics that relate to polyomaviruses sero-positivity in Chinese populations. An increasing sero-prevalence of JCV and MCV with age was found among Chinese or US populations and may be explained by a combination of continuous transmission of the pathogen throughout life and persistent antibody responses.

### H. pylori

Previous studies have reported a substantial regional variation in *H. pylori* prevalence globally with China having a relatively high prevalence. In a meta-analysis of 670 572 participants from 98 Chinese studies during 1983–2018, using different methods to measure *H. pylori*, 49.6% overall (ranging from 20.6% to 81.8%) were sero-positive for *H. pylori*.[28] The present study also observed a significantly higher age-standardised sero-prevalence in China than UK (50.5% vs 34.1%) using multiplex serology (online supplemental table 4). Within China, regional differences in *H. pylori* sero-prevalence have not been previously explored and we found substantial variation across the 10 study areas, which may be attributed partly to levels of socioeconomic development and sanitation, as *H. pylori* is commonly transmitted person-to-person by saliva, bouts of gastroenteritis within households and/or spread by faecal contamination of food or water.[29 30] There is a good evidence that a combination of untreated water, crowded conditions and poor hygiene contributes to higher *H. pylori* prevalence in developing countries.[28 31] However, we were not able to explore these factors comprehensively due to the limited information available, but, as in many other populations, we observed a declining or plateaued sero-prevalence of *H. pylori* by birth cohort.[32] Sero-positivity for *H. pylori* was less

common in participants with a history of peptic ulcer, which may reflect reverse causation as individuals with peptic ulcer would be treated by antibiotics to eradicate *H. pylori*, leading to lower sero-prevalence.

### C. trachomatis

*C. trachomatis* is the most commonly diagnosed bacterial sexually transmitted infection (STI) worldwide. Since *C. trachomatis* has not yet been included as a reportable STI in the national STI surveillance programme in China, detailed epidemiological data are lacking. Previous population-based studies reported a *C. trachomatis* current infection prevalence of 2.1%–4.1% in Chinese,[6] while in the present study which measured current and past infection status prevalence was 47.6% overall, which was higher in older participants likely due to cumulative exposure of the infection. Although both HHV-2 and *C. trachomatis* are sexually transmitted, there was a fivefold difference in sero-prevalence between these two pathogens in CKB (8.3% vs 44.1%), which was not evident in the UKB population (16.8% vs 23.1%). This is unexplained and warrants further investigation. Individuals with higher socioeconomic status were less likely to be infected, which was also reported in a national stratified probability sample of 3426 Chinese (aged 20 to 64 years, 49.3% women) in 1999–2000.[6]

### T. gondii

*T. gondii* is a zoonotic parasite infecting all warm-blooded animals. Usually, infected humans are asymptomatic, but in pregnant women, the infection may lead to abortion, stillbirth or other serious consequences in newborns. Based on two national serological surveys in 15 provinces in China, the ELISA-based sero-prevalence of *T. gondii* increased from 5.2% in 1988–1992 to 7.9% in 2001–2004.[33 34] A recent serological survey involving 2634 healthy individuals in China showed a 12.5% anti-*T. gondii* IgG-positive rate,[35] which is comparable to our estimate (14.2%). Consistent with previous findings in China, we found the sero-prevalence was highest in Liuzhou (18.4%), Guangxi province, where the consumption of raw or undercooked meat is popular among local residents, which is an important factor in the prevalence of food-borne parasitic diseases.[36]

### Strengths and limitations

The main strength of the present study is the simultaneous assessment of antibodies against multiple pathogens, using a validated and custom designed multiplex serology platform. Moreover, the broad age range of participants also allowed assessment of differences in the pathogens' sero-prevalence across the different birth cohorts and/or age groups, in addition to associations with socioeconomic and lifestyle factors. Nevertheless, the study also has limitations. These findings may not be generalisable to general populations in China. Geographic variation in the sero-prevalence of pathogens was observed among the 10 diverse regions of China that were included in the CKB

study. However, with only ~200 participants from each region, the sero-prevalence in that area may not be accurately estimated. The multiplex serology measurements are designed to detect cumulative (ie, past and present) exposure in a high-throughput setting and may partially explain the high sero-prevalence of several pathogens compared with previous estimates.[10–12] Furthermore, the information collected about possible determinants or correlates for certain infections was limited, particularly in relation to sexual behaviours. Finally, the number of participants tested did not allow us to explore prospectively the associations of the sero-prevalence of multiple pathogens with the incidence of cancers and other NCDs.

## Conclusions

In summary, among Chinese adults, the burden of pathogen infections was high and varied by area, sex, birth cohort and certain socioeconomic and lifestyle factors. While some infections may represent past or transient infections with little or no major health consequences, others may represent chronic and persistent infections, which may lead to increased risk of certain cancers and many other NCDs. The development of a custom-designed multiplex panel and its application in CKB should provide a reliable basis for future well-powered and prospective investigations of the associations between chronic infections and consequent disease outcomes in the Chinese population.

**Author affiliations**
[1]Clinical Trial Service Unit & Epidemiological Studies Unit (CTSU), Nuffield Department of Population Health, University of Oxford, Oxford, UK
[2]Medical Research Council Population Health Research Unit (MRC PHRU), Nuffield Department of Population Health, University of Oxford, Oxford, UK
[3]The Wellcome Centre for Human Genetics, University of Oxford, Oxford, UK
[4]Infections and Cancer Epidemiology Division, German Cancer Research Center, Heidelberg, Germany
[5]Fuwai Hospital Chinese Academy of Medical Sciences, National Center for Cardiovascular Diseases, Beijing, China
[6]Department of Epidemiology & Biostatistics, School of Public Health, Peking University Health Science Center, Beijing, China
[7]Department of Statistics, University of Warwick, Coventry, UK
[8]Early Detection, Prevention and Infections Branch, International Agency for Research on Cancer, Lyon, France
[9]Department of Epidemiology and Biostatistics, Peking University School of Public Health, Beijing, China

**Contributors** PY, IM, CK, LY and ZC had full access to the data in the study and take responsibility for the integrity of the data and the accuracy of the data analysis (as the guarantors). RJ, JB and TW conducted the assay measurement in the DKFZ Laboratory. Statistical analysis: PY. Drafting of the manuscript: PY. Acquisition, analysis, or interpretation of data: PY, IM, TW, LY and ZC. Critical revision of the manuscript for important intellectual content: PY, IM, CK, AM, RJ, JB, MP, CM, GC. Administrative, technical or material support: NA, YG, YC, RW, JL, CY, and LL. Supervision: TW, LY and ZC. TW, LY and ZC contributed equally to this report.

**Funding** This work was supported by CRUK PRC-Project Award (C56488/A24504). The China Kadoorie Biobank (CKB) baseline survey was supported by the Kadoorie Charitable Foundation, Hong Kong. CKB long-term follow-up was supported by grants from the UK Wellcome Trust (212946/Z/18/Z, 202922/Z/16/Z, 104085/Z/14/Z, 088158/Z/09/Z), National Natural Science Foundation of China (91843302), and National Key Research and Development Program of China (2016YFC 0900500, 0900501, 0900504, 1303904), The UK Medical Research Council (MC_UU_00017/1, MC_UU_12026/2 MC_U137686851), Cancer Research UK (C16077/A29186; C500/A16896) and British Heart Foundation

(CH/1996001/9454). The chief acknowledgement is to the participants, the project staff, and the China National Centre for Disease Control and Prevention (CDC) and its regional offices for access to death and disease registries. The Chinese National Health Insurance scheme provides electronic linkage to all hospitalised individuals.

**Map disclaimer** The inclusion of any map (including the depiction of any boundaries therein), or of any geographic or locational reference, does not imply the expression of any opinion whatsoever on the part of BMJ concerning the legal status of any country, territory, jurisdiction or area or of its authorities. Any such expression remains solely that of the relevant source and is not endorsed by BMJ. Maps are provided without any warranty of any kind, either express or implied.

**Competing interests** None declared.

**Patient and public involvement** Patients and/or the public were not involved in the design, or conduct, or reporting, or dissemination plans of this research.

**Patient consent for publication** Consent obtained directly from patient(s)

**Ethics approval** Ethics approval was obtained from Oxford Tropical Research Ethics Committee (Ref: 025-04), and the Chinese Centre for Disease Control and Prevention Ethical Review Committee (005/2004). Participants gave informed consent to participate in the study before taking part.

**Provenance and peer review** Not commissioned; externally peer reviewed.

**Data availability statement** Data are available upon reasonable request. Anonymised baseline, resurvey and cause-specific mortality and morbidity data are available for access through a formal application in CKB website (www.ckbiobank. org). The application will then be reviewed by a Data Access Committee. Further details about access policy and procedures can be found online at www.ckbiobank. org.

**ORCID iDs**
Pang Yao http://orcid.org/0000-0002-6875-7435
Rima Jeske http://orcid.org/0000-0003-4596-3778
Jun Lv http://orcid.org/0000-0001-7916-3870

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
