## [Reviewer comments · BMJ Open]

ARTICLE DETAILS

TITLE (PROVISIONAL)	Sero-prevalence of 19 infectious pathogens and associated factors among middle-aged and elderly Chinese adults: a cross-sectional study
AUTHORS	Yao, Pang; Millwood, Iona; Kartsonaki, Christiana; Mentzer, Alexander J; Allen, Naomi; Jeske, Rima; Butt, Julia; Guo, Yu; Chen, Yiping; Walters, Robin; Lv, Jun; Yu, Canqing; Plummer, Martyn; de Martel, Catherine; Clifford, Gary; LI, Li-ming; Waterboer, Tim; Yang, Ling; Chen, Zhengming

VERSION 1 – REVIEW

REVIEWER	Kong, Yuanyuan Beijing Friendship Hospital
REVIEW RETURNED	01-Dec-2021

GENERAL COMMENTS	This paper attempted to investigate the sero-prevalence of 19 infectious pathogens and its associated factors in 10 areas (5 urban and 5 rural) of China. Some issues regarding to study design and casual inference of this study remained to be clarified. I hope the authors find the following suggestions and comments be helpful to further improve the quality of this paper. Major comments: 1. In page 5 line 40, the authors mentioned that “The present study population was a sub-cohort of 2000 participants from a case-subcohort study of gastric and oesophageal cancers in China Kadoorie Biobank (CKB).” The study population could not well represent the general populations in China. Though Table 1 showed the baseline characteristics were similar between this subcohort and the overall CKB cohort, however, the variables listed in Table 1 may not be sufficient enough to describe the homogeneity or heterogeneity. Did authors consider and discuss the generalization of the findings in this study? In addition, please explain why the inclusion criteria was defined as “those alive with no history of cancer 2 years after entering the study”, did the authors mean the gastric and oesophageal cancers occurred two years later after the subjects enrolled in the CKB study? If so, what would be the potential influence when include people who have cancer within two years after their enrollment in the original study?2. In page 8 line 115, the authors stated that “No difference by sex were observed for most pathogens.” However, there were no “p-values” provided in Table 2. The Authors should provide statistical results to support the statement of “no difference”.3. In Result part, the present study found that the HBV infection rate was approximately 44%, which is much higher than the results reported in several Chinese nationwide surveys. This again may relate to the issue on representative of sampling in this study. Authors also mentioned these nationwide surveys of HBV in
---

	China. What is the possible explanation for this finding? I suggest the authors should discuss this finding in more details. 4. In page 12 line 195, the present study showed that “participants with reported prior cirrhosis or chronic hepatitis had >7-fold higher risk of HBV seropositivity.” Actually, HBV infection is the main pathogen of chronic hepatitis B and HBV-related cirrhosis. Cirrhosis or chronic hepatitis cannot be the risk factors of HBV infection. I suggest the authors should be more careful on the casual inference by using a cross-sectional study design. 5. This study concluded that “these infections may contribute to increased risk of certain cancers and other non-communicable chronic diseases.” However, the results in the present study cannot support these conclusions. 6. There was no environmental factors analysis in this study. I suggest the authors should keep consistent in the study aims, results analysis and discussion. 7. The authors analyzed the correlations between different pathogens, and the results were shown in Correlation matrix figures (corplot). The non-significant correlation coefficients should be represented by blank boxes in correlation matrix. The detailed p-values and correlation coefficients should be provided in Tables. Minor comments:  1. All the font “p” of p-value and font “r” of correlation coefficients should be italic fonts. 2. Please double check the format consistency of references. Several references contain the published online date.
--	---

REVIEWER	Wu, Jiang Beijing Center for Disease Prevention and Control, Institute of Immunization and Vaccine
REVIEW RETURNED	07-Dec-2021

GENERAL COMMENTS	1: Adults usually define 18 years old and above, while the age range of the population in this paper refers to 30-79. It is inconsistent with the title and will mislead some conclusions. For example, Since the beginning of the 1990s, mass vaccination campaigns have had a major impact on reducing Hepatitis B Virus (HBV) infection in China. The authors also cited other studies showing this. Accordingly, it is estimated that the people age of 18-30 are historically in that period. Therefore, the authors cannot extrapolate HBV seroprevalence in adults from this study, although they believe that the age range prevalence in this study is consistent with previous studies. 2: Some studies have demonstrated nonspecific binding of sera in bead-based immunoassay, resulting in high nonspecific background. How to validate the agent and quality control of the multiplex serology platform. Usually, for a newly established detection method, it is necessary to verify its sensitivity, specificity, and accuracy. Therefore, the author should add the validation data of the multiplex serological detection method used in the article, illustrating the reliability of their data and results. For example, there have been commercially available HBV tests on the market for decades. How do the results compare to theirs? 3: The author did not mention how the respective MFI values of the 19 antigen-specific cutoff values were determined. And line 79, No references provided.
--

	4: The author should explain in detail about the sample method, and the reason why rural samples collected from provinces and urban samples collected from cities. For example, the author did not describe the number recruited in each site (Figure S1). Are there enough samples to reflect the sero-prevalence in each site? Although the author claimed that the study design had been previously reported, this paper is independent and self-contained, so it should be explained in detail.
--	--

VERSION 1 – AUTHOR RESPONSE

Reviewer: 1

Dr. Yuanyuan Kong, Beijing Friendship Hospital Comments to the Author:

This paper attempted to investigate the sero-prevalence of 19 infectious pathogens and its associated factors in 10 areas (5 urban and 5 rural) of China. Some issues regarding to study design and casual inference of this study remained to be clarified. I hope the authors find the following suggestions and comments be helpful to further improve the quality of this paper.

Major comments:

1. In page 5 line 40, the authors mentioned that “The present study population was a sub-cohort of 2000 participants from a case-subcohort study of gastric and oesophageal cancers in China Kadoorie Biobank (CKB).” The study population could not well represent the general populations in China. Though Table 1 showed the baseline characteristics were similar between this subcohort and the overall CKB cohort, however, the variables listed in Table 1 may not be sufficient enough to describe the homogeneity or heterogeneity. Did authors consider and discuss the generalization of the findings in this study? In addition, please explain why the inclusion criteria was defined as “those alive with no history of cancer 2 years after entering the study”, did the authors mean the gastric and oesophageal cancers occurred two years later after the subjects enrolled in the CKB study? If so, what would be the potential influence when include people who have cancer within two years after their enrollment in the original study?

Response: The study population in this study is a subcohort of CKB participants from a case-cohort study, which aimed to explore the role of H. pylori and other pathogens in gastric cancer development. To avoid the reverse causality and also the reduced antibody level from precancerous lesions, e.g. gastric atrophy, we excluded all participants with history of cancer within two years after study entry from the case-cohort study. The sub-cohort participants were well representative of the entire CKB cohort which included >0.5 million Chinese adults aged 30-79 from 10 diverse regions (as shown in Table 1), which was semi-representative of the Chinese adult population. Therefore, the sero-prevalence of pathogens from our subcohort participants reflects, to some extent, the current infection profile in China. We have now clarified the study design and added further discussion about the generalization issue in the text (Page 18, Line 338-342).

1. In page 8 line 115, the authors stated that “No difference by sex were observed for most pathogens.” However, there were no “p-values” provided in Table 2. The Authors should provide statistical results to support the statement of “no difference”.

Response: Done. Statistical results have now been presented in Table 2 (Page 26).

1. In Result part, the present study found that the HBV infection rate was approximately 44%, which is much higher than the results reported in several Chinese nationwide surveys. This again may relate to the issue on representative of sampling in this study. Authors also mentioned these nationwide surveys of HBV in China. What is the possible explanation for this finding? I suggest the authors should discuss this finding in more details.

Response: The HBV markers measured in this study are antibody responses to HBc and HBe antigens, which indicate past as well as current infection, not to HBsAg which reflects chronic infection. As described in Discussion (Page 12, Line 199-205), our sero-prevalence estimate was lower than previously published studies that also measured the same antigens (HBc and HBe).

1. In page 12 line 195, the present study showed that “participants with reported prior cirrhosis or chronic hepatitis had >7-fold higher risk of HBV seropositivity.” Actually, HBV infection is the main pathogen of chronic hepatitis B and HBV-related cirrhosis. Cirrhosis or chronic hepatitis cannot be the risk factors of HBV infection. I suggest the authors should be more careful on the casual inference by using a cross-sectional study design.

Response: Done. We have now amended the results as follows (Page 13, Line 214-216):

“Our study showed that HBV sero-positivity was >7-fold higher in participants who had reported prior cirrhosis or chronic hepatitis than in their counterparties.”

1. This study concluded that “these infections may contribute to increased risk of certain cancers and other non-communicable chronic diseases.” However, the results in the present study cannot support these conclusions.

Response: We have revised the concluding sentence to “These infections may contribute to the burden of certain cancers and other non-communicable diseases.” (Page 2, Line 23-25).

1. There was no environmental factors analysis in this study. I suggest the authors should keep consistent in the study aims, results analysis and discussion.

Response: Done. We have removed environmental factors throughout the manuscript accordingly.

1. The authors analyzed the correlations between different pathogens, and the results were shown in Correlation matrix figures (corplot). The non-significant correlation coefficients should be represented by blank boxes in correlation matrix. The detailed p-values and correlation coefficients should be provided in Tables.

Response: We have added p values (Page 11, Line 168-172). The correlation matrix follows a standard presentation format.

Minor comments:

1. All the font “p” of p-value and font “r” of correlation coefficients should be italic fonts.

Response: Done

1. Please double check the format consistency of references. Several references contain the published online date.

Response: Done

Reviewer: 2

Dr. Jiang Wu, Beijing Center for Disease Prevention and Control Comments to the Author:

1. Adults usually define 18 years old and above, while the age range of the population in this paper refers to 30-79. It is inconsistent with the title and will mislead some conclusions. For example, Since the beginning of the 1990s, mass vaccination campaigns have had a major impact on reducing Hepatitis B Virus (HBV) infection in China. The authors also cited other studies showing this. Accordingly, it is estimated that the people age of 18-30 are historically in that period. Therefore, the authors cannot extrapolate HBV seroprevalence in adults from this study, although they believe that the age range prevalence in this study is consistent with previous studies.

Response: We have amended the title as follows:

“Sero-prevalence of 19 infectious pathogens and associated factors among middle-aged and elderly Chinese adults: a cross-sectional study”

1. Some studies have demonstrated nonspecific binding of sera in bead-based immunoassay, resulting in high nonspecific background. How to validate the agent and quality control of the multiplex serology platform. Usually, for a newly established detection method, it is necessary to verify its sensitivity, specificity, and accuracy. Therefore, the author should add the validation data of the multiplex serological detection method used in the article, illustrating the

reliability of their data and results. For example, there have been commercially available HBV tests on the market for decades. How do the results compare to theirs?

Response: The present study used a custom-designed and validated multiplex serology panel, based on Luminex fluorescent bead-based flow cytometry technology. The DKFZ laboratory has developed a serology portfolio including over 400 antigens for a range of viruses and bacteria, which have been comprehensively validated against respective gold-standard serological assays.¹⁻³ The DKFZ laboratory have performed extensive assay optimisation and quality control (for example, screening for background inhibitors it was found that serum pre-incubation with polyvinylalcohol, polyvinylpyrrolidone and a proprietary reagent significantly reduced non-specific background.⁴

1. The author did not mention how the respective MFI values of the 19 antigen-specific cutoff values were determined. And line 79, No references provided.

Response: We have added antigen-specific cutoff determination in Methods (Page 7, Line 84-85) as follows:

“Antigen-specific cut-offs for *H. pylori* antigens were applied as described previously,¹⁵ and were quality assured by the visual inflection point method.¹⁶”

1. The author should explain in detail about the sample method, and the reason why rural samples collected from provinces and urban samples collected from cities. For example, the author did not describe the number recruited in each site (Figure S1). Are there enough samples to reflect the sero-prevalence in each site? Although the author claimed that the study design had been previously reported, this paper is independent and self-contained, so it should be explained in detail.

Response: These province/city names are the conventional area labels used in the CKB study, and further details on the recruitment sites are provided on the CKB website (<https://www.ckbiobank.org>). We have added the study design and data collection methods in the Methods (Page 7, Line 38-43) and supplementary file, the number of participants from each region in the legend of Figure 2 (Page 24), and the low number in each region was included as a limitation in the Discussion as follows (Page 18, Line 341-342):

“However, with only ~200 participants from each region, the sero-prevalence in that area may not be accurately estimated.”

1

Reference

1. Brenner N, Mentzer AJ, Butt J, et al. Validation of Multiplex Serology detecting human herpesviruses 1-5. *PLoS One* 2018;13(12):e0209379-e79. doi: 10.1371/journal.pone.0209379
2. Waterboer T, Sehr P, Michael KM, et al. Multiplex Human Papillomavirus serology based on in situ-purified glutathione S-transferase fusion proteins. *Clinical Chemistry* 2005;51(10):1845-53. doi: 10.1373/clinchem.2005.052381
3. Brenner N, Mentzer AJ, Butt J, et al. Validation of Multiplex Serology for human hepatitis viruses B and C, human T-lymphotropic virus 1 and *Toxoplasma gondii*. *PLoS One* 2019;14(1):e0210407. doi: 10.1371/journal.pone.0210407
4. Waterboer T, Sehr P, Pawlita M. Suppression of non-specific binding in serological Luminex assays. *J Immunol Methods* 2006;309(1-2):200-4. doi: 10.1016/j.jim.2005.11.008 [published Online First: 2006/01/13]